# Decoding natural image stimuli from fMRI data with a surface-based convolutional network

**Zijin Gu**[1]                                                                    ZG243@CORNELL.EDU
**Keith Jamison**[2]                                                        KWJ2001@MED.CORNELL.EDU
**Amy Kuceyeski**[*2]                                                    AMK2012@MED.CORNELL.EDU
**Mert Sabuncu**[*1,2]                                                    MSABUNCU@CORNELL.EDU

[1] *School of Electrical and Computer Engineering, Cornell University and Cornell Tech, New York, New York, USA*

[2] *Department of Radiology, Weill Cornell Medicine, New York, New York, USA*

**Editors:** Accepted for publication at MIDL 2023

## Abstract

Due to the low signal-to-noise ratio and limited resolution of functional MRI data, and the high complexity of natural images, reconstructing a visual stimulus from human brain fMRI measurements is a challenging task. In this work, we propose a novel approach for this task, which we call Cortex2Image, to decode visual stimuli with high semantic fidelity and rich fine-grained detail. In particular, we train a surface-based convolutional network model that maps from brain response to semantic image features first (Cortex2Semantic). We then combine this model with a high-quality image generator (Instance-Conditioned GAN) to train another mapping from brain response to fine-grained image features using a variational approach (Cortex2Detail). Image reconstructions obtained by our proposed method achieve state-of-the-art semantic fidelity, while yielding good fine-grained similarity with the ground-truth stimulus. Our code is available on https://github.com/zijin-gu/meshconv-decoding.git.

**Keywords:** functional MRI, neural decoding, image reconstruction

## 1. Introduction

Neural decoding, especially visual decoding that reconstructs the external visual stimulus from brain activity patterns, is a challenging task in neuroscience research. Development of neuroimaging techniques, e.g. functional magnetic resonance imaging (fMRI), and breakthroughs in artificial intelligence like deep learning, have been recently used to build models that can map brain responses to visual stimulus with high quality, thus paving the way for future "mind-reading" technologies that can have clinical and scientific applications.

**Previous work**   Previous studies have demonstrated the feasibility of mapping from brain responses to visual stimulus. The predominant approach in early works used linear models to map fMRI-derived measurements to hand-crafted image features, such as Gabor wavelet coefficients (Miyawaki et al., 2008; Naselaris et al., 2009; Nishimoto et al., 2011). More recently, deep neural networks (DNNs) have become popular to compute nonlinear mappings from brain signals to visual stimuli. In these DNN methods, Generative Adversarial

---

[*] Contributed equally

Networks (GANs) have become a popular building block to reconstruct realistic images (St-Yves and Naselaris, 2018; Seeliger et al., 2018; Shen et al., 2019a; Lin et al., 2019; Shen et al., 2019b; VanRullen and Reddy, 2019; Fang et al., 2020; Gu et al., 2022; Ozcelik et al., 2022), and self-supervision techniques, e.g. via an autoencoder (AE), have shown to improve generalization (Du et al., 2017; Beliy et al., 2019; Fang et al., 2020). While several stages of optimization may be required, these approaches have been able to reconstruct images with good similarity to the viewed ground-truth images.

Virtually all previous work on visual decoding handles fMRI data by selecting voxels that belong to certain visual regions-of-interest (ROIs), e.g. early visual areas V1 - V4 (Ress and Heeger, 2003; Gardner et al., 2005), or higher visual areas FFA (Kanwisher et al., 1997), PPA (Epstein and Kanwisher, 1998) and LOC (Malach et al., 1995). These voxel measurements are then flattened into 1D vectors that are in turn provided as input to the visual decoder. This approach has several important weaknesses: 1) ROI definitions and the selection involved in fMRI preprocessing are subjective and can vary across individuals; and 2) the spatial topology of the 2D cortical areas is largely ignored when using the vectorized voxel responses.

In this work, we present a novel visual decoding framework to address these weaknesses. The proposed Cortex2Image framework contains one Cortex2Semantic model, one Cortex2Detail model and one pre-trained and frozen image generator called Instance-Conditioned GAN (IC-GAN). Comparing with the most state-of-the-art method which uses ridge regression to separately predict the feature and noise input vectors to IC-GAN (Ozcelik et al., 2022), our method has several key improvements: 1) the proposed Cortex2Image model has an architecture shared across individual subjects that consumes cortex-wide brain activity, since it uses a standardized mesh representation of the cortex, instead of relying on specific ROIs; 2) surface convolutions used in our model can exploit spatial information in brain activity patterns; 3) our approach to train the Cortex2Detail model in an end-to-end fashion by combining with IC-GAN improves computational efficiency comparing with previous methods which need extra steps on the noise vector optimization. Our experiments demonstrate that the proposed visual decoding framework is able to produce high-fidelity natural-looking images that achieve state-of-the-art accuracy in matching high-level semantics, while capturing much of the image details.

## 2. Method

### 2.1. Dataset

**Natural Scenes Dataset** We used a recently collected densely-sampled fMRI dataset, called Natural Scenes Dataset (NSD) (Allen et al., 2021) to train and test the proposed framework [1]. In short, the NSD contains natural image stimuli and whole brain responses for 8 participants collected over approximately a year. Among the 10,000 images each participant viewed, 1,000 images are shared across individuals while the remaining 9,000 images are mutually exclusive across subjects. We used the data from four participants who completed all the designed trials (10,000 images in 30,000 trials).

---

1. http://naturalscenesdataset.org/

## 2.2. Cortex2Image: mapping brain responses to images

The Cortex2Image model consists of three components: a Cortex2Semantic model, a Cortex2Detail model and a image generator from IC-GAN, see Figure 1a.

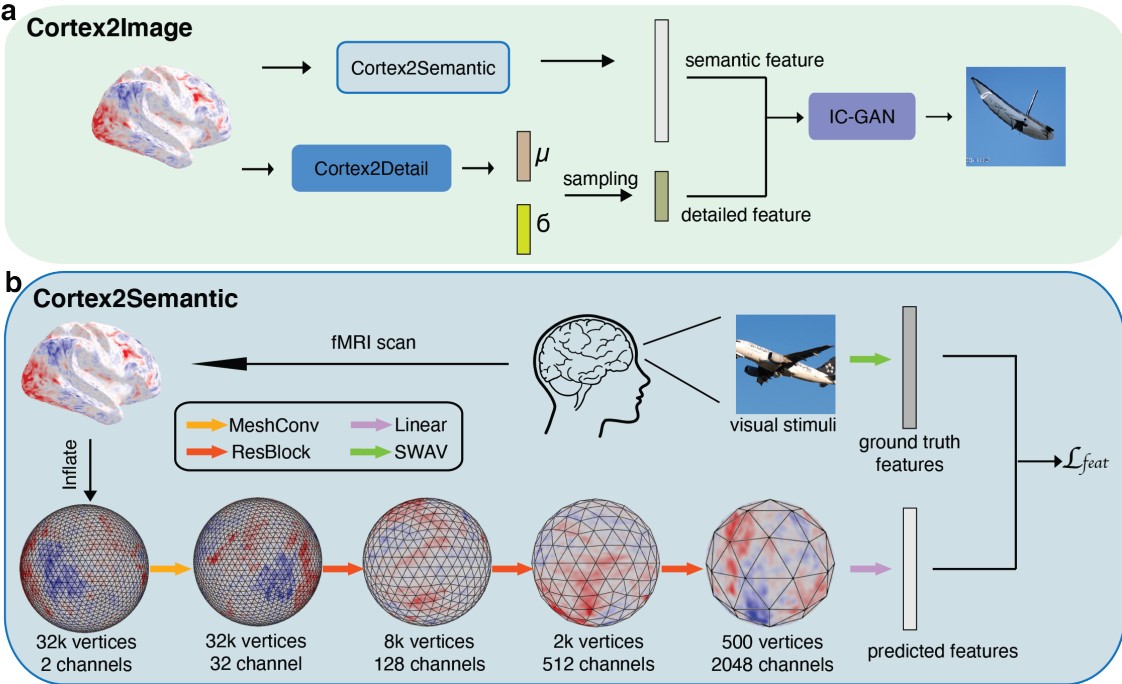

Figure 1: (a) The Cortex2Image model architecture , which consists of a Cortex2Semantic model (detailed in panel b), a Cortex2Detail model that shares the same archiecture as Cortex2Semantic model, and an image generator from IC-GAN.

**Cortex2Semantic**   The Cortex2Semantic model maps the fMRI brain response to the image semantic features, e.g. object categories, corresponding to that response. Instead of flattening the brain voxel responses and simply applying a linear model like most previous works (Ozcelik et al., 2022), we employ a spherical convolutional network using spherical kernels (Jiang et al., 2019) which takes into account the neighborhood information on the cortical surface (Ngo et al., 2022). In our implementation, brain voxel responses are represented as multi-channel data attached to the icosahedral mesh vertices, with each channel corresponding to a hemisphere. The proposed Cortex2Semantic model, shown in Figure 1b, contains a mesh convolution layer, followed by three downstream residual blocks and a dense layer. The loss function is the $l_2$ distance between the predicted semantic vector and ground truth semantic vector, which is an image representation extracted by a fixed pretrained network called "Swapping The unsupervised Assignments between Views (SwAV)" (Caron et al., 2020). SwAV is trained in self-supervised way and is also used as the fea-

ture extractor in IC-GAN training (Casanova et al., 2021). SwAV features are used in the IC-GAN image generator that we describe below.

**Cortex2Detail** In addition to the semantic image features, we trained another Cortex2Detail model which maps brain responses to fine-grained (detail) image features that capture factors like color, size, and orientation. This model's output is a stochastic layer that represents an approximate posterior distribution on the fine-grained image details and is trained with a variational approach. The architecture is the same as Cortex2Semantic, except for the final layer where the dimensions are different.

**IC-GAN Generator** In IC-GAN (Casanova et al., 2021), a complex distribution on, say images, $p(\boldsymbol{x})$ is approximated by learning simpler conditional distributions around data points: i.e. $p(\boldsymbol{x}) \approx \frac{1}{M} \sum_i p(\boldsymbol{x}|\boldsymbol{h}_i)$, where $M$ is the number of data samples and $\boldsymbol{h}_i$ is the features of data sample $\boldsymbol{x}_i$. The instance features are extracted using some embedding function. In IC-GAN, this embedding function is the pre-trained SwAV model (Caron et al., 2020) that computes semantic image features. The conditional distributions $p(\boldsymbol{x}|\boldsymbol{h}_i)$ are modeled implicitly using a generative network $G_\theta$ parameterized by $\theta$. The generator takes in a random vector sampled from a standard Gaussian prior $\boldsymbol{z} \sim N(0, I)$ and a feature vector $\boldsymbol{h}_i$ corresponding to an image $\boldsymbol{x}_i$ sampled from a dataset; and generates a random sample $\boldsymbol{x}$ which is conditioned on $\boldsymbol{h}_i$. The generator is jointly trained with a discriminator in an adversarial approach. One advantage of IC-GAN is its capability of transferring to unseen datasets during training meanwhile keeping high fidelity. Note that the random vector $\boldsymbol{z}$ can be regarded as capturing fine-grained image details like color, size, and orientation, whereas $\boldsymbol{h}_i$ reflects semantics. Therefore, the generated images will share the same semantic meaning as the image that is conditioned on, but will vary in details.

The complete Cortex2Image model is shown in Figure 1a. We emphasize that the semantic image features can be pre-computed for all ground-truth images using the pre-trained SwAV model. The Cortex2Semantic model, in turn, is trained directly on these features. On the other hand, obtaining the ground-truth fine-grained (detail) feature vector for a given image involves a costly optimization process, which we describe below. For training Cortex2Detail, we opted to avoid this computationally expensive step. Instead, we train Cortex2Detail in conjunction with of the Cortex2Image model in an end-to-end fashion, where both the Cortex2Semantic and Cortex2Detail model outputs are served as input to the IC-GAN generator to compute an image reconstruction. Here we take a variational, instead of a deterministic approach in predicting the detail vector based on the assumption that the exact details of an image cannot be uniquely decoded from noisy fMRI data. To ensure the reconstructed image has both high semantic and detail fidelity, the loss is composed of three parts:

$$\begin{aligned} L = &- \lambda_{recon} \mathbb{E}_{z \sim q(z|x_i)}[\log p_\theta(x_i|z)] \\ &+ \lambda_{KL} \mathbb{KL}(q(z|y_i)||p(z)) \\ &- \lambda_{feat} \mathbb{E}_{z \sim q(z|x_i)}[\log f_\phi(x_i|z)] \end{aligned}$$

where the first term is the pixel level reconstruction loss, the second term is the Kullback-Leibler divergence between the approximate posterior on the fine-grained detail vector and its prior, which is a standard Gaussian, and the third term is the SwAV latent space feature loss.

## 2.3. Baselines and upper bound

We compare our model with a state-of-the-art visual decoding framework (Ozcelik et al., 2022), where the authors used a linear model with $l_2$ regularization. Similar to this model, we fitted two ridge regression models with the flattened fMRI data as the input: one for predicting the semantic image features and the other for the fine-grained image details. Same as Cortex2Semantic, the ground truth semantic features are derived using the pretrained SWAV latent space. To obtain the "ground truth" fine-grained details, we implemented the following iterative optimization procedure. We inputted the ground truth semantic image features and a randomly initialized fine-grained detail vector to the IC-GAN generator and optimized over the detail vector using the covariance matrix adaptation evolution strategy (CMAES) (Hansen and Ostermeier, 2001). This global optimization strategy works better than local optimization strategies, e.g. gradient-based methods. The loss function is the $l_2$ distance between the reconstructed image and the ground truth image in SwAV latent space. Note that this process is computationally demanding, e.g. on average 50 seconds for one optimization and around 140 hours for one subject if not done in parallel, and can be unstable (Ozcelik et al., 2022).

We compare our full Cortex2Image models with the following models:

1) RidgeSemantic: images reconstructed using a linear ridge regression approach that predicts the semantic vectors, combined with randomly sampled detail vectors from $N \sim (0, 1)$.

2) RidgeImage: images reconstructed using a linear ridge regression approach that predicts both the semantic and optimized fine-grained detail vectors. This method is computationally demanding, due to expensive step of estimating the fine-grained detail vectors for all the training images.

3) Cortex2Semantic: images reconstructed using the proposed Cortext2Semantic approach that predicts the semantic vectors, combined with randomly sampled detail vectors from $N \sim (0, 1)$.

RidgeSemantic and Cortex2Semantic (which is an ablation of the proposed method) models were considered as baselines because it has been reported that reconstructions from predicted semantic features and randomly sampled fine-grained detail vectors can achieve better results than from predicted fine-grained detail vectors (Ozcelik et al., 2022).

The synthetic upper bound is the best reconstruction that the IC-GAN generator can achieve, given the semantic feature vector of the presented ground-truth stimulus. This is computed via the optimization of the fine-grained detail vector, as described above. Note that there is no fMRI data involved in this reconstruction.

## 2.4. Evaluation metrics

Model evaluation was done qualitatively and quantitatively. For qualitative evaluation, we displayed the reconstructed image along with the ground truth image to compare them visually. For quantitative evaluation, we adopted both high-level and low-level (fine-grained) metrics. High-level metrics include the latent space distance of SWAV and EfficientNet-B1 (Tan and Le, 2019). EfficientNet-B1 was chosen as it ranks top on BrainScore (Schrimpf et al., 2020), which measures the similarity between artificial neural networks and the brain's

mechanisms for core object recognition. Low-level metrics include the classical Structural Similarity (SSIM) and pixel-wise correlation.

## 2.5. Implementation Details

We implemented the Cortex2Image model using PyTorch. The input surface meshes to the Cortex2Semantic and Cortex2Detail models are the left and right FreeSurfer cortical meshes (Van Essen et al., 2012) with 32,492 vertices per brain hemisphere. The downstream residual blocks reduce the number of vertices on the mesh from 32,492 to 92, with 7,842, 2,562 and 492 in between, and the number of channels increases from 2 to 2048, with 32, 128, 512 in between. The mesh convolution has stride 1 and all other Conv1d layers have kernel size 1 and stride 1. For the Cortex2Semantic model, the last linear layer maps the averaged pooled features to the final output of size 2048. For Cortex2Detail model, there are two penultimate fully connected layers, one for the mean and the other for the variance of the multivariate Gaussian vector and both map the averaged pooled features to the output of 119. The image generator from IC-GAN is built upon Big-GAN architecture (Brock et al., 2018) with the class embedding layer replaced by a feature embedding layer. The output images are of resolution $256 \times 256$. SwAV pre-trained model with ResNet-50 model (Caron et al., 2020) is used as the instance feature extractor, as it has been recently shown that unsupervised models are able to achieve equal or better performance than supervised models by learning hierarchical features to capture the structure of brain responses across the ventral visual stream (Konkle and Alvarez, 2022; Zhuang et al., 2021). The Cortex2Semantic model is trained first and then kept fixed during the training of Cortex2Detail. The IC-GAN generator is also fixed during training. The baseline linear models were implemented using the Ridge regression model implemented in the sklearn python package. Ridge models have about twice as many parameters as the Cortex models.

We trained subject-specific models using their own data given the unique response properties of individuals. The shared 1,000 images were used as the held-out test set, and the remaining 9,000 samples were divided into training set (8,500) and validation sets (500). AdamW optimizer was employed for training the neural networks. Learning rate was set to $1e - 3$ for Cortex2Semantic and $1e - 4$ for Cortex2Detail. Weight decay was set to 0.1. For the training of Cortex2Detail, $\lambda_{recon}$, $\lambda_{KL}$ and $\lambda_{feat}$ were set to $1e - 6$, $1e - 8$ and 1, respectively, which were observed to give best performance on the validation set. Batch size was 8. The maximum training epoch was set to 1000, while monitoring the validation loss to save the model that gave the best performance. All the quantitative results reported are based on applying the final model to the 1000 test response-image samples.

## 3. Results

In Figure 2 we visualize the viewed stimulus (top row, black frames) along with the synthetic upper bound reconstructions from extracted semantic vectors and optimized detail vectors (second row, blue frames), and the reconstructions from four subjects' fMRI data (rows 3-6, green frames). We provide both well and poorly decoded examples in different image categories, including animals, buildings, food, etc. Looking at the well decoded examples, the fMRI reconstructions from the proposed method are able to largely match the semantic contents of the viewed images, for example, a zebra on the grass, a train on the railway

Figure 2: Image reconstruction from fMRI data from four individuals using the NSD dataset.

track and a plane in the sky. Though fine-grained details (like color, pose and exact object shape/borders) are more difficult to recover, some details are preserved, such as the black and white stripes on the zebra body, the color of the pizza, the location of the tower, the orientation of the train and airplane, the framing of the athletes bodies, and the proportion of ground vs sky in the landscape.

There are still some failures of the proposed method, as shown in the poorly decoded examples. We noticed that these viewed images usually have multiple objects of different categories, and objects or their predictive features are occluded or only partially presented. For example, for a viewed image of a person and a giraffe, some subjects' models only reconstructed the person while others reconstructed the animal. The reason could be the different focuses of the subjects during fMRI and the limitation of the image generator. Future work can add eyetracking data to examine attention's effect on the reconstructed images. It should be noted that we have observed that subject-specific models do not generalize well across subjects (results not shown) and don't yield meaningful results. This is likely due to the variable response properties of subjects and the different signal-to-noise ratio levels in the fMRI data.

### 3.1. Cortex2Image vs RidgeImage

Figure 3 provides a comparison of our Cortex2Image reconstructions (green frame) with the reconstructions from RidgeImage baseline (grey frame). By leveraging the spatial information within the brain data, our reconstruction visually shares more similarities with the ground truth stimulus in both high-level (semantic) and low-level (fine-grained) features. The decoded object classes are correct in most cases, as well as the color and position of the objects and background. The RidgeImage model is able to reconstruct some of the semantic features, however, the overall visual quality is poor. In our quantitative comparisons, our reconstructions achieve better results than the RidgeImage baseline in terms of all evaluation metrics, except SSIM.

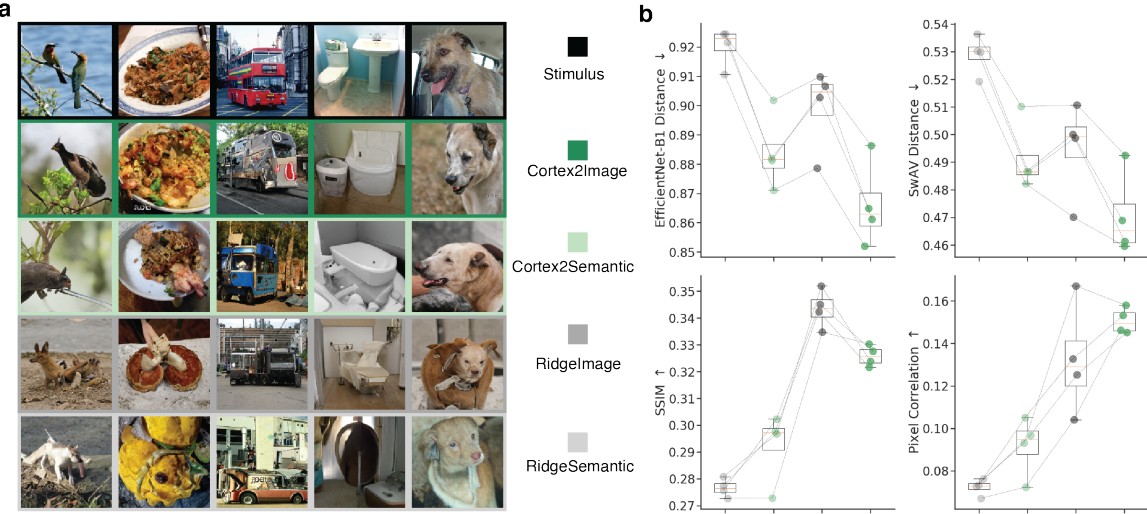

Figure 3: (a) Qualitative and (b) quantitative comparisons of image reconstruction results for the four different decoding models. For each metric, ↓ means lower is better and ↑ means higher is better.

## 3.2. The effect of considering low level features in the decoding models

We characterized the significance of capturing fine-grained details in our model architecture. In this vein, we first compared the reconstructed images from our full Cortex2Image model with the images reconstructed with the Cortex2Semantic model, see Figure 3. We observe that the Cortex2Semantic model is able to capture object category, but the Cortex2Image model can achieve significantly better results in predicting the fine-grained features, such as the orientation of the bus and the arrangement of the toilet and washbasin. Similarly, when comparing the reconstructions of RidgeSemantic (light grey frame) and RidgeImage (grey frame), we observe a boost in quality that is due to the model's consideration of fine-grained details in the decoding. This is also reflected in the quantitative results, presented in Figure 3b, where we see that the Cortex2Image model outperforms Cortex2Semantic model and the RidgeImage model is better than the RidgeSemantic model in all metrics.

## 4. Conclusion

Our work highlights the importance of modeling the spatial information in fMRI brain responses for visual decoding. Our results show that the proposed surface-based convolutional architecture can achieve superior quality in image reconstructions over current state-of-the-art baseline models. Additionally, combining the image generator in model training greatly reduces the time and computation cost when obtaining the "ground truth" detail vector, as noted in previous work. Overall, the present work is a novel approach for decoding visual stimuli from brain activity and has potential in brain machine interfaces and studying human brain response properties via external stimuli alterations.

## Acknowledgments

This work was funded by the following grants: NIH RF1 MH123232 and Cornell/Weill Cornell Intercampus Pilot Grant.

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

## Appendix A. Details of the Natural Scenes Dataset

Over the course of 30-40 7T fMRI scans with whole-brain gradient-echo EPI, 1.8-mm iso-voxel and 1.6s TR, each subject viewed 9,000–10,000 color natural scenes in 22,000 to 30,000 trials. Images were sourced from the Microsoft Common Objects in Context (COCO) database (Lin et al., 2014), and square cropped to present at a size of $8.4° \times 8.4°$. Images were presented for 3s on and 1s off while subjects were asked to fixate centrally. In order to encourage maintenance of attention, they also performed a long-term continuous recognition task on the images. Temporal interpolation and spatial interpolation were used to preprocess the fMRI data to correct for slice time differences and head motion. Then a general linear model (GLM) was applied to estimate the single-trial beta weights which represented the voxel-wise response to the image presented. Cortical surface-based response maps were generated using FreeSurfer [2].

---

2. [http://surfer.nmr.mgh.harvard.edu/](http://surfer.nmr.mgh.harvard.edu/)

## Appendix B. More examples on the effect of Cortex2Detail

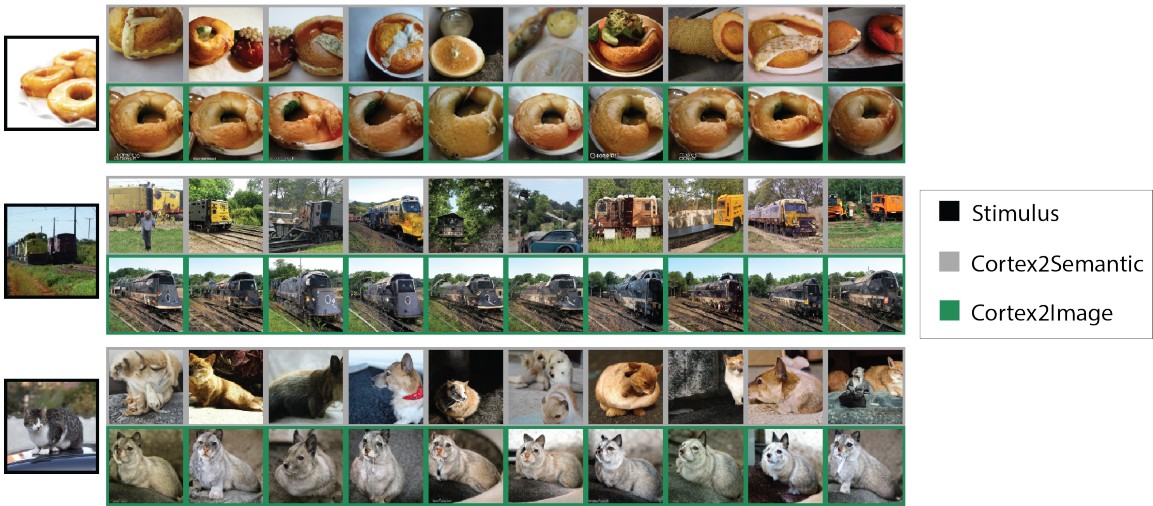

Figure 4: Qualitative comparison of image reconstructions from Cortex2Image and Cortex2Semantic, with random sampling.

For the donuts example, the random sampling reconstructs many dessert/food-like images, which can be bread, cakes or oranges, while the Cortex2Image reconstructions all contain a clear hole in the middle of a donut which exactly matches the shape of the original image.

