# OpenReview forum: "Decoding natural image stimuli from fMRI data with a surface-based convolutional network"
_MIDL.io/2023/Conference — MIDL 2023 Oral_

### Official Review · Reviewer_otk3 · 2023-02-03

**Confidence:** 5
**Preliminary Rating:** 4
**Recommendation:** Poster

**Summary:**


The manuscript proposes a method to reconstruct stimulation images from fMRI 7T data. The method is composed by 3 blocks: a. Cortex2Semantic, mapping the fMRI brain response to the image semantic features, b. Cortex2Detail, mapping fMRI to fine-grained (detail) image features, and c. IC-GAN a generator which makes images. Four subject-specific models were trained, and visual results were shown.


**Strengths:**

The visual results are very good, probably the best in the literature. The approach of working with activation surfaces rather than volumetric data is original and gives good results. The architecture is clever and I look forward to reading the journal version!

**Weaknesses:**

a.	The proposed model comparisons do not fully convince me. If they count as an ablation study, which does not look exhausting, then not sure if there is a real comparison with state-of-the-art methods.
b.	Very nice results in Figure 2, but it would increase the transparency if this Figure would be divided into two sub-figures with good and bad results.
c.	It is unclear how Cortex2Semantic and Cortex2Detail, having similar architecture and input, will produce representations on semantic features and fine-grained (detail) image features, respectively. Some steps are missing in the description.
d.	It would be interesting to see inter-subjects results. I agree that this leads to many problems, but it’s such an important target that a discussion and some considerations (at least) are needed.
e.	Despite I like the idea of using activation meshes as input data, we also need to consider that many volumetric data work as well. It would be nice to have a comparison or at least a discussion of what a similar work could be.


**Deanonymize Review:**

yes

**Detailed Comments:**

1.	I agree that many past methods relied on flattening the brain voxel responses into 1D vectors, but many examples of methods don’t, so they keep spatial consistency. Examples could come from Michal Irani’s group. I would adjust the Previous work section accordingly.
2.	Page 5 states “RidgeFeature and Cortex2Semantic (which is an ablation of the proposed method)”. What RidgeFeature is?
3.	Conclusion, I would remove the sub-sentence “which has been largely ignored in the literature.” I don’t think the field ignores the problem, since many groups publish several papers. Maybe it’s not mainstream, but definitely not ignored.
4.	Conclusion, I foresee the long-term goal of “helping us to understand how humans perceive the world around them.”. But I don’t see it very relevant here. Maybe lower the sentence a bit or be more precise.
5.	It would be interesting to specify the number of parameters of the two main nets.
6.	It’s not clear the mapping between Cortex2Index and the noise vector. Please provide more details on the framework architecture.


**Paper Type:**

methodological development

**Questions To Address In The Rebuttal:**

The main weakness is the lack of description of the two components Cortex2Semantic and Cortex2Detail. I would like to see results supporting representation specialisation on semantic features and fine-grained features. Section 3.2 doesn’t exhaustively answer the question.

At this point, the model does not tell us much about the brain. Also considering the spatial resolution of the 7T data, can we provide more insights on the visual reconstruction process? For example, highlight which brain areas contribute more to the final visual reconstruction. A classical experiment could be forcing to zero specific brain area activations and try to measure the performance drop.

---

### Official Review · Reviewer_KQRY · 2023-02-03

**Confidence:** 4
**Preliminary Rating:** 5
**Recommendation:** Oral, Poster

**Summary:**

This paper proposes a new deep learning framework named Cortex2Image for decoding a visual stimulus with fine grained detail from fMRI measurement data while preserving semantic fidelity. Specifically, Cortex2Image consists of two portions, Cortex2Semantic - which is a surface based convolutional network that maps brain responses to semantic features, and Cortex2Detail- a generative adversarial network to map to fine grained features and trained using a variational approach.

Evaluation is performed on the natural scenes dataset (NSD) against ridge regression approaches, an ablation on the framework and a synthetic upper bound (images generated by the GAN given ground truth semantic features). The results for the method compare favorably in terms of qualitative and quantitative performance (computed via proxy similarity measures).

**Strengths:**

1. The presentation of the paper is very good. The methodology is clearly explained and well motivated. The baseline comparisons and experiments have been carefully chosen such that they are relevant to examining the main claims and propositions. Although the comparative metrics are on the subjective side, the framework demonstrates consistent improvements.

2. The methodological novelty is a major strength of the paper. The design choices are sound, clearly explained, and adequately justified. The approach offers an interesting and innovative data-driven solution to a rather challenging problem in brain activity decoding.

**Weaknesses:**

Given that the framework relies on a GAN for image generation, it is unclear what the performance is on examples where there is poor agreement in terms of the quantitative measure (larger distances, poor SSIM). Does the model hallucinate irrelevant content here, are the finer details lost, or both? Adding a couple of illustrative examples and some discussion on this point could be helpful. It would also be useful to know what the quantitative results are for the recovered examples that are reasonable (such as those presented in the appendix and results)

**Deanonymize Review:**

no

**Detailed Comments:**

Other comments:

1. The color correspondences in Fig 3 are very hard to parse in a paper pdf. A suggestion would be to add the method labels on the left corresponding to each of the rows.

2. Please increase the font size for text in all of the Figures, they are hard to read.

3. This is probably beyond the scope of this submission- have the authors considered having human annotators flag the relevance/quality of recovered examples, given the stimulus? This may provide a better indication of the relative performances of the framework and baselines.

**Paper Type:**

methodological development

**Questions To Address In The Rebuttal:**

Referring to the comments in the weaknesses section, adding examples where the method performs poorly (on the adopted metrics) could help improve the presentation of the results and round out the discussion of the methodology. It would be interesting to provide some insight and discussion as to what applications such frameworks could be used for in practice.

---

### Official Review · Reviewer_KejX · 2023-02-06

**Confidence:** 4
**Preliminary Rating:** 5
**Recommendation:** Best Paper Award, Oral

**Summary:**

The authors develop a model to reconstruct visual stimuli from fmri-derived cortical surface maps.
It is mostly an applications paper, adapting an existing Spherical CNN, but the overall model architecture is an interesting approach.
Their method aims to reproduce the pretrained encoded latent vector instead of the actual image, and splits learning fine vs coarse details, with the former learned through a variational approach.

Results show that the reproduced images although not perfect (which is expected) captures both categorical details as well as some fine details.

Comparison is made with a state of the art ridge regression baseline.

**Strengths:**

The analysis is done very well.  There is an ablation-like study of the results with and without fine details for both the proposed method and the ridge regression baseline. The results with (Cortex2Image) and without (Cortex2Semantic) fine detail are convincing enough by inspection.
There is numerical comparison with the ground truth, and a recognition of the 'upper bound' limitation.

The paper is well written, with more space dedicated to explaining the more important details.

The results seem remarkable (comparing to the baseline in the paper and similar recent work)

Generally, deep modelling of fMRI is a topic worthy of focus.

**Weaknesses:**

The quality seems to vary a little between subjects, but this is not addressed.
For example, subject 4's  zebra reconstruction is neither striped nor very equine. subject 3's 'pizza' is not immediately recognisable as such, etc.  This is somewhat expected since the models are trained and inferred on a subject-by-subject basis. Can the authors offer any insight as to whether the model might have worked across subjects?

**Deanonymize Review:**

no

**Detailed Comments:**




**Paper Type:**

both

**Questions To Address In The Rebuttal:**

The paper is very good and I strongly support this paper's inclusion in MIDL 2023.
I have recommended it for an oral as I believe that the topic would be of audience interest.
Therefore, there is nothing for the authors to address.

---

### Meta-Review · Area_Chair_LrZC · 2023-02-23

**Recommendation:** Accept (Oral)
**Confidence:** 5

**Metareview:**

This is a very strong paper implementing a surface based decoding method from rich 7T fMRI data. The method is novel, well validated and performs well relative to baselines. All reviewers agree that it is an excellent paper, likely to generate lively discussion at the conference and authors improved further during rebuttal.